# Greater Consumption of Total and Individual Lignans and Dietary Fibers Were Significantly Associated with Lowered Risk of Hip Fracture—A 1:1 Matched Case–Control Study among Chinese Elderly Men and Women

**DOI:** 10.3390/nu14051100

**Published:** 2022-03-05

**Authors:** Zhaomin Liu, Bailing Chen, Baolin Li, Cheng Wang, Guoyi Li, Wenting Cao, Fangfang Zeng, Yuming Chen

**Affiliations:** 1Guangdong Provincial Key Laboratory of Food, Nutrition and Health, School of Public Health, Sun Yat-sen University, Guangzhou 510080, China; liuzhm8@mail.sysu.edu.cn (Z.L.); wangcheng@mail.sysu.edu.cn (C.W.); ligy26@mail2.sysu.edu.cn (G.L.); caowenting107@163.com (W.C.); 2Department of Bone Surgery, The 1st Affiliated Hospital of Sun Yat-sen University, Guangzhou 510080, China; chenbl2012@163.com; 3Guangzhou Orthopaedics Trauma Hospital, Guangzhou 510045, China; baolin197488@163.com; 4Department of Epidemiology, School of Public Health, Hainan Medical University, Haikou 571199, China; 5Department of Epidemiology, School of Basic Medical Science, Jinan University, Guangzhou 510623, China; zengffjnu@126.com; 6Department of Epidemiology, School of Public Health, Sun Yat-sen University, Guangzhou 510080, China

**Keywords:** lignans, hip fracture, Chinese older adults, case–control study

## Abstract

The study aims to examine the association of dietary intake of lignans with the risk of hip fractures in Chinese older adults. This was a 1:1 age- and gender- matched case–control study. Dietary survey was conducted by face-to-face interviews using a 79-item validated food frequency questionnaire. Habitual intake of total and individual lignans (matairesinol, secoisolariciresinol, pinoresinol, and lariciresinol) was estimated based on the available lignans databases. Conditional logistic regression was used to examine the relationship of dietary total and individual lignans, lignan-rich foods (vegetables, fruits, nuts, and cereals) and dietary fibers with the risk of hip fracture. A total of 1070 pairs of hip fracture incident cases and controls were recruited. Compared with the lowest quartile, the highest quartile group showed a reduced hip fracture risk by 76.3% (0.237, 95% CI: 0.103–0.544, P_trend_ < 0.001) for total lignans, and 62.5% (0.375, 95% CI: 0.194–0.724, P_trend_ = 0.001) for dietary fibers. Similar findings were observed for individual lignans, the estimated enterolactone level, as well as lignans from vegetables and nuts. We concluded that greater consumption of total and individual lignans, and lignan-rich foods were significantly associated with decreased risk of hip fracture.

## 1. Introduction

Hip fractures are the leading cause of osteoporotic-related disability and death. With the rapid increase of elderly population and their extended life expectancy, cases of hip fractures are predicted to rise from 1.7 million in 1990 to 6.3 million in 2050 worldwide [1]. The age-specific prevalence of hip fracture has been declining in most western countries, but still been increasing in some areas of Asia [2]. Exploration of an effective strategy for hip fracture prevention would have essential implications for public health and reduce the health resources burden in Asian regions.

Diet, as a modifiable factor, plays an important role on bone health. Phytoestrogens have a structural similarity to endogenous estrogens and behave as both selective estrogen receptor modulator and antioxidant agents, suggesting a possible role in improvement of bone health. Epidemiological evidence suggested that phytoestrogen intake could reduce bone loss and fracture risk [3]. The major groups of dietary phytoestrogens include isoflavones, lignans, and coumestans. In contrast to isoflavones being the most consumed phytoestrogen in Asian population, lignans are considered as the major source of phytoestrogen in the western diet [4]. Lignans are biphenolic compounds, occurring as glycosides in the fiber layer of plants, and are found in high amounts in whole grains, fruits, vegetables, and nuts. The major subclasses of lignans are matairesinol (MAT), secoisolariciresinol (SECO), pinoresinol (PINO), and lariciresinol (LARI). In mammals, plant lignans can be converted into bioactive enterolignans (enterolactone and enterodiol) by the intestinal microflora [5].

Although in vitro studies and animal experiments have shown that lignans could confer benefits on bone health by suppressing osteoclast formation via modulating RANKL signaling pathways, or activation of estrogen receptor (ER)-α/-β through actions on both osteoblasts and osteoclasts, and thus increase bone mass [6,7], epidemiological studies are limited and often retrospective in nature, reported inconsistent findings. Cross-sectional studies in postmenopausal women suggested that increased enterolactone excretion was associated with reduced rate of bone loss [8] or increased bone mineral density (BMD) [9]. Several clinical trials have testified the effectiveness of flaxseed lignans on bone measures; however, most of them were conducted among postmenopausal women and did not report significant impacts of lignans on BMD, or markers of bone metabolism [10,11,12]. Observational studies on lignans have been mainly conducted in European or American countries. Asians ingested a smaller amount of dietary lignans than westerns (200 to around 300 vs. >1000 μg/d) [4]. In addition, China is in a transitional period from the plant-food-dominated dietary pattern of decades ago to the current animal foods pattern [13]. It is uncertain whether such a low consumption of dietary lignans is favorable for bone health. In addition, the outcomes of most of the observational studies on phytoestrogens and bone health have primarily addressed BMD, but rarely in fracture incidence. In fact, low BMD status can only explain part of fracture risk [14]. To date, no studies have evaluated the association of habitual lignans consumption and risk of fractures among the Asian population. We thus aimed to explore the association of dietary lignans intakes with the risk of hip fracture based on a case–control study among Chinese older adults. We hypothesized that hip fracture was associated with low intakes of lignans. Since lignans are highly correlated with dietary fibers, and different subtypes of lignans might exert differing impacts on bone health, we thus further tested the associations of fracture risk with individual lignans (subtypes), lignan-rich food sources (cereals, vegetables, fruits, and nuts), and dietary fibers to substantiate the findings.

## 2. Materials and Methods

This was a 1:1 age- (±3 years) and gender-matched case–control study conducted between June 2009 and 2015 in Guangdong, China. The original and main purpose was to explore and test the essential dietary and environmental factors related to hip fractures in Chinese older adults. The study protocol has been approved by the Ethics Committee of *** (omitted here for the double-blind review process). All of the participants have signed the informed consent forms before enrollment.

The detailed inclusion and exclusion criteria and methodologies have been reported previously [15]. Briefly, eligible cases were those who were newly diagnosed (within 2 weeks) with a hip fracture by X-ray; were aged from 55 to 80 years; had been living in Guangdong Province for more than 10 years. Patients were recruited from four hospitals in Guangdong Province. Patients were excluded if they had pathological or high-energy fractures; had dietary habits changed dramatically within the previous 5 years; had poor vision; or reported certain chronic diseases which might influence dietary intakes or routine physical activities. A total of 1070 patients were recruited, among which 691 (58.9%) were femoral neck fractures, and 355 (30.2%) were intertrochanteric fractures. For each of cases, we enrolled 1:1 age- (±3 years) and gender-matched controls, which applied the same criteria with cases, except for the new incidence of hip fracture. The controls were recruited from either community (82.3%) or hospital (17.7%) through a variety of strategies such as written invitations, flyers, or referrals.

Individual information was collected by trained interviewers through face-to-face interviews using a structured questionnaire on socio-demographics, lifestyle habits, family history of chronic diseases, medical histories, and dietary habits. Dietary intake was assessed by a 79-item validated food frequency questionnaire (FFQ) [16]. Of these, 57 items contributed lignans values. The FFQ was conducted based on habitual food intake during the year preceding the fracture diagnosis (for case patients) or interview (for control participants). Participants were asked to report their food consumption frequency using 5 predefined categories (never, per day, per week, per month, or per year). A commonly used portion size was specified for each food items (e.g., bowl, slice, glass, or unit). Photographs of the foods and their corresponding portion sizes were provided for the participants as visual aids to quantify the food portions. Dietary energy and nutrients intakes were calculated according to the Chinese Food Composition Table 2004 and 2009. The validity and reproducibility of the FFQ have been confirmed among the local population by six three-day dietary records over a one-year period [16].

Total dietary lignans intake was estimated based on several published database of lignans [17,18,19,20] which included the four most prevalent dietary precursors of enterolactone (SECO, MAT, LARI, and PINO). Intakes of total lignans were estimated by summing the amounts of four precursors of lignans. The major lignans database was from Canada [18], and supplemented by Japanese [19] and Dutch data [17] as well as an online database—Phenol-Explorer [20]. We assigned lignans values for each food item in the FFQ according to the following protocol: (1) lignans values in the database were converted to mg/100 g food and expressed on a wet weight basis; (2) if the FFQ listed several kinds of foods in the same line of food item, we preferentially used the average lignans values for several most commonly consumed foods; (3) for an individual food item, if there was no lignans value available in the database, a value of a similar food item of lignans data was assigned, otherwise the value of 0 was assigned. Daily intakes of total and individual lignans were determined by multiplying the daily consumption amounts of individual food items by their respective lignans content. As the activity of the intestinal microbiota influences the metabolism of dietary lignans into enterolignans [21,22], we further calculated the expected amount of mammalian lignans, the enterolactone based on the experimental results using an established formula [5]. The estimated bioavailable enterolactone (ENL, μg/100 g) = 0.62 × MAT + 0.72 × SECO + 1.01 × LARI + 0.55 × PINO.

Statistical analyses were performed using IBM Statistic SPSS software (version 20.0) with two-tailed *p* < 0.05 as statistical significance. Data were analyzed using Student’s *t* test for continuous variables and chi-square test for categorical variables. The energy-adjusted dietary total and individual lignans were categorized into quartiles (Q1–Q4) on the basis of the distribution by genders among the controls. Multivariate conditional logistic regression was used to estimate the odds ratios (ORs) and 95% confidence intervals (CIs) for the associations between dietary factors and the risk of hip fracture, with the lowest quartile (Q1) as the reference group. Tests for linear trends were performed using the categories of quartile of dietary variables as continuous variables. Subgroup analyses were conducted to examine the potential effect modifications by subgroup variables such as gender (men or women), sources of controls (hospital or community), BMI (<24 or ≥24), healthy lifestyle scores (below or above median), regular calcium supplementation (yes or no), and habitual tea-drinking (yes or no). Sensitivity analyses were also performed to testify if the associations were remained after exclusion of participants of current smoking, alcohol drinking, and type 2 diabetes. Since dietary lignans and fibers were highly correlated, additional sensitivity analysis with inclusion of both two variables in the same regression model resulted in an alleviated magnitude and statistical non-significance in association with hip fracture risk. There was a significant interaction between dietary lignans and fibers quartiles, suggesting a possible synergistic effect between them (data not shown).

## 3. Results

### 3.1. Participants Characteristics

The demographics and selected risk factors related to fractures among the 1070 pairs cases and controls (mean age of 70.6 ± 7.2 y) are shown in Table 1. Compared with the controls, the fracture patients had a lower BMI, household income, and education attainment above high school, engaged in less total physical activity, were less likely to have regular calcium and multivitamin supplementation and drink tea, had a lower prevalence of medical history of hyperlipidemia and coronary heart disease, but higher rate of stroke and previous fractures, with a higher experience of active and passive smoking.

### 3.2. Dietary Intakes by Cases and Controls

Dietary intakes of cases and controls are shown in Table 2. Case patients had significant lower intake of fruits and vegetables, fish, milk, and dairy products (*p* < 0.001), lower intake of total energy (*p* = 0.002), energy-adjusted soy protein (*p* = 0.030), dietary fat (*p* = 0.033), and calcium intake (*p* = 0.001), compared with controls. The average daily intakes of total lignans were 414.5 ± 1197.6 and 525.1 ± 418.4 μg/d for cases and controls, respectively (*p* < 0.001). The major dietary sources of total lignans were vegetables (56.7%), nuts (22.8%), cereals (9.6%), and fruits (3.6%) (Appendix A).

### 3.3. Associations of Dietary Lignans Consumption and Hip Fracture Risks

Multivariate logistic regression indicated that the highest quartile of total lignans was associated with reduced risk of hip fracture by 76.3% (OR: 0.237; 95% CI: 0.103–0.544; P_trend_ < 0.001) compared with the lowest quartile. Similar findings were observed for individual lignans as well as the estimated ENL level (Table 3). Except for SECO (P_trend_ = 0.084), other subtypes of lignans demonstrated a significant dose–response manner across the quartiles (P_trend_ < 0.001).

For the lignans-rich food sources and dietary fibers (Table 4), comparing the extreme quartiles, the highest quartile group had a significant fracture risk reduction by 53.0% (OR: 0.470; 95%CI: 0.231~0.954; P_trend_ = 0.012) for lignans from vegetables, 75.5% (OR: 0.245; 95%CI: 0.115~0.521; P_trend_ < 0.001) for lignans from nuts and 62.5% (OR: 0.375; 95%CI: 0.194~0.724; P_trend_ < 0.001) for dietary fibers. No significant risk alterations were observed for lignans from cereals and fruits.

Further subgroup analyses (Table 5) indicated similarly inverse associations between total lignans intake and hip fracture risk among men and women, habitual tea-drinking (yes or no), and overall healthy lifestyle scores (high or low). The favorable associations were more evident in participants with community controls, less obesity (BMI < 24), and who were under no regular calcium supplementation. Sensitivity analyses with the exclusion of participants who engage in smoking, drink alcohol regularly, and currently have diabetes did not materially change the conclusions.

## 4. Discussion

This case–control study among Chinese older adults revealed a significantly inverse and dose-responsive association between dietary intakes of total and individual lignans, lignans from vegetables and nuts, and dietary fibers with the risk of hip fracture. To our knowledge, this is the first case–control study conducted among Chinese older adults to evaluate the association of dietary lignans intake with hip fracture risk. The major strengths of this study were its 1:1 age- and gender-matched case–control design with a relatively large sample size and multicenter data collection; its outcome of fracture incidence not the surrogate bone measures; the availability of detailed dietary data and extensive lifestyle information, as well as the wide variance of dietary lignans intake in this population.

Different types of phytoestrogens may have different effects on organisms, depending on the target tissues and the type of activated receptors [23]. Phytoestrogens have a selective affinity to estrogen receptors and selectively increased the transcriptional activity of the estrogen response element or activation of second messenger pathways [24]. Phytoestrogens might also cause adverse health effects by acting as endocrine disruptors, which may result in the differences in their metabolism and consequent bioavailability [25]. Compared with the western population, Asians usually have much lower lignans but higher isoflavones intake (18.7 mg/d in current study) [4]. Our findings indicated that even with a low intake, habitual lignans consumption was still notably associated with lowered hip fracture risk in an obvious dose–response manner even after adjustment for dietary isoflavones and fibers intakes. Except for the properties as both antioxidant and estrogenic phytoestrogens, the synergistic effects of lignans with other bone-protective compounds (such as isoflavones, fibers, or unsaturated fatty acids, etc.) in plant foods may partly explain the positive findings. In addition, high lignans intake might be an indicator of a healthy lifestyle [26]. Our analysis also showed that participants with higher lignans intakes had healthier lifestyle profiles such as lower BMI, higher percentage of tea-drinking, regular calcium and multivitamin supplementation, and being more physical active while having less sedentary time. However, the stratification analysis by high and low healthy lifestyle scores, or adjustment for several lifestyle factors in multivariate models did not significantly alter the associations, suggesting an independent association between lignans intake and fracture risk.

In this study, we not only observed a favorable effect of total and individual lignans intake, but also lignans from vegetables and nuts on reduced hip fracture risk. To our knowledge, this is the first epidemiological study to explore the relationship of lignans from nuts with hip fracture risk in older adults. Our findings are supported by several observational studies, which reported that greater vegetables intake were associated with reduced bone turnover [27], better bone mass, and lowered risk of hip fracture [28]. According to the lignans database, lignans were rich in nuts, soybeans, whole-grain bread, and vegetables (green onion, green pepper, and cabbages). It was thus implied that older adults with high risk of fracture should be encouraged to include more lignan-rich foods as part of their habitual diet. Our findings from univariate regression even suggested an increased hip fracture risk by lignans from total cereals intake, although this association was attenuated to non-significance after full adjustment of potential confounders. Evidence suggested a favorable association of whole grains intake with improved bone health due to the highest level of lignans in whole grains relative to other cereals [29]. However, our dietary data were limited to fully investigate the whole cereals consumption. Animal experiments have suggested that dietary fibers could promote calcium absorption [30], improve bone turnover, or increase bone mass [31]. Data from human controlled trials also demonstrated that dietary fibers increase calcium absorption and retention, and modulate bone mineralization in either children/adolescents [32] or postmenopausal women [33,34,35]. The cohort of Framingham Offspring Study also indicated that higher dietary fiber could modestly reduce hip bone loss in men [36]. To our knowledge, no human studies have examined the associations of fiber intake with risk of fracture. The possible biological mechanisms for the beneficial effect of dietary fiber on the skeleton could be due to the prebiotic properties of fiber in modulating the microbial composition and releasing short-chain fatty acids during fermentation in the gut to improve calcium absorption [37], reduce inflammation, and stimulate hormones to regulate bone metabolism [38].

There are several mechanisms underlying the putative beneficial effects of lignans on bone health. As one major phytoestrogen in the western diet, lignans possess both antioxidant property and weak estrogenic activity. As a strong antioxidant agent, lignans may counteract inflammation and exert protection against bone loss. The estrogenic effects of lignans are widely assumed to be mediated by the mammalian enterolignans, especially enterolactone via activating estrogen receptors (ERs) and lead to subsequent ER-mediated gene transcription in the regulation of hormonal responsiveness [39]. Lignans may interfere with estrogen metabolism by alteration of serum hormone-binding globulins, estradiol, or estrogen metabolites, which could influence tissue estrogen exposure and therefore bone health [12]. In addition, in vitro and animal experiments have shown that lignans can increase plasma insulin-like growth factor-1 (IGF-1) and insulin-like growth factor binding protein-3 (IGFBP-3), as well as estrogen levels, which are relevant to bone metabolism [40].

The limitations of this study should be acknowledged. Case–control studies are vulnerable to selection and recall bias, reversal causality, and residual confounding. First, 17% of controls in our study were recruited from hospitals, which may lead to selection bias. However, the hospital controls were recruited under certain conditions that had no apparent associations with dietary modifications. Second, reversal causality could not be fully excluded in a case–control study, because the dietary data were collected after diagnosis. We minimized this possibility by interviewing cases as soon as possible after their diagnosis and meticulously excluding individuals with essential changes in their diet habits in the last five years.

Third, FFQ are known to contain a certain degree of measurement error, although it has been validated. FFQ provide a well-established means for capturing long-term dietary behavior for most chronic diseases. We attempted to minimize the recall bias and improve accuracy of reporting through use of food photographs with usual intake portions to assist participants with quantification of intake; use of personal interviews not self-administration. In addition, the FFQ was not specifically designed for collection of dietary information on lignans intake and we did not investigate flaxseed consumption, which is the richest source of dietary lignans; however, more than 72% food items in our FFQ contribute lignans values and the Chinese population had much lower flaxseed but higher sesame consumption than westerners. Furthermore, we divided the cohort into quartile categories of intake, which reduces the influence of extreme data points. We additionally explored the associations of lignan-rich foods and dietary fibers on hip fracture risk and observed similar findings.

Fourth, as in most observational studies, we didn’t quantify the biomarkers for lignans exposure due to resource limitation. However, serum or urinary enterolignans can only reflect a short-term intake due to their relatively short half-life (4.4 h for enterodiol and 12.6 h for enterolactone) and may not take into account variability over time [41]. Previous studies have reported that even a semi-quantitative FFQ is a valid indicator of the usual intake of phytoestrogens relative to multiple 24-h dietary recalls and multiple plasma samples [42]. In addition, at population level, a clear time-integrated dose–response relationship exists between lignans consumption and concentrations of enterolignans in plasma or urine [41]. Even the misclassification of the exposure in a case–control study design can only tend to be non-differential and lead to attenuation of possible association. Another limitation could be that the lignans databases we used in the current study were from different countries, since Chinese data were unavailable. The major lignans database in our analysis were from Japan where food varieties and dietary patterns are similar to those of the Chinese population.

Finally, possible residual confounding by unknown factors cannot be entirely excluded even if we have adjusted a variety of possible confounders in the multivariable regression models. Our findings need confirmation in large-scale prospective studies. As well, sample size for some subgroup analyses of interest was limited, and might have led to some minor inconsistencies.

## 5. Conclusions

Our case–control study among Chinese elderly men and women indicated that greater intake of dietary total and individual lignans, lignans from vegetables and nuts, as well as dietary fibers were associated with a lowered risk of hip fracture. Further prospective studies and investigations on enterolignans measured in serum or urine with fracture risk are warranted to confirm our findings and to disentangle the possible mechanisms involved. As dietary intake of lignans is modifiable, this finding would be of public health importance with respect to fracture prevention in Asian older adults.

## Figures and Tables

**Table 1 nutrients-14-01100-t001:** Demographics and selected risk factors of fracture by cases and controls in Chinese elderly men and women of Guangzhou, China.

Characteristics	Cases	Controls	*p*
*n*	1070	1070	
Men/women	277/793	276/794	0.961
Age (year)	70.7 ± 7.3	70.5 ± 7.0	0.326
BMI (kg/m^2^)	21.8 ± 6.4	23.2 ± 3.2	<0.001
Education (≥high school, *n*%)	335 (31.5%)	561 (52.4%)	<0.001
Retired (*n*, %)	691 (83.6%)	768 (92.4%)	<0.001
Living in rural area (%)	405 (48.8%)	419 (50.4%)	0.494
Household income (≥3000, *n*%)	210 (19.8%)	398 (37.2%)	<0.001
Married or cohabitated (%)	671 (63.6%)	791 (74.0%)	<0.001
Orientation of house (facing to the sun, *n* %)	740 (80.4%)	723 (78.5%)	0.305
Depression (%)	282 (34.1%)	301 (36.2%)	0.421
Smoking (*n* %)			
Current smoking (*n* %)	180 (16.8%)	135 (12.6%)	0.006
Passive smoking (*n* %)	226 (21.2%)	172 (16.1%)	0.003
Habitual alcohol drinking (*n* %)	73 (6.8%)	77 (7.2%)	0.739
Habitual tea-drinking (*n* %)	371 (34.7%)	540 (50.5%)	<0.001
Regular calcium entation (*n* %)	313 (29.3%)	455 (42.5%)	<0.001
Regular multivitamin usage (*n* %)	102 (9.5%)	307 (28.7%)	<0.001
Medical history (*n* %)			
Hypertension	328 (30.7%)	319 (29.8%)	0.373
Hyperlipidemia	58 (5.4%)	265 (24.8%)	<0.001
Diabetes	101 (9.4%)	75 (7.0%)	0.115
CHD	47 (4.4%)	96 (9.0%)	<0.001
Stroke	52 (4.9%)	20 (1.9%)	<0.001
Previous fracture	60 (18.1%)	150 (7.7%)	<0.001
Family history of fracture	136 (13.3%)	142 (12.7%)	0.681
Fall in past two years	260 (24.3%)	236 (22.1%)	0.214
Total physical activity (Mets)	67.7 ± 40.2	80.9 ± 52.1	<0.001
Sports (Mets)	1.14 ± 2.55	2.37 ± 3.38	<0.001
Sedentary time (hours)	6.6 ± 2.7	6.2 ± 2.6	0.015

Data were expressed as means ± standard deviation for continuous variables and *n* (%) for categorical variables. *p* values for *t*-test (continuous variables) and Chi-square test (categorical variables) were indicated. BMI: body mass index; CHD: coronary heart disease; Mets: Metabolic equivalent of task, the energy expenditure per min per kg body weight.

**Table 2 nutrients-14-01100-t002:** Dietary intakes of foods and nutrients among cases patients and controls of Chinese elderly men and women, Guangzhou, China.

Foods, Nutrients and Lignans	Cases (*n* = 1070)	Controls (*n* = 1070)	*p*
Food groups (g/d)			
Cereals	609.8 ± 268.6	599.5 ± 210.2	0.326
Vegetables	247.3 ± 137.2	321.8 ± 175.9	<0.001
Fruits	68.0 ± 64.4	98.9 ± 79.0	<0.001
Legumes	86.4 ± 433.3	72.3 ± 292.7	0.379
Red meat	74.5 ± 74.3	76.9 ± 55.9	0.403
Total fish	29.9 ± 30.8	39.6 ± 39.2	<0.001
Milk and dairy products	68.0 ± 114.6	87.9 ± 116.2	<0.001
Energy and nutrients intake *			
Energy (kcal/d)	1479.4 ± 953.3	1582.6 ± 468.3	0.002
Protein (g/d)	72.7 ± 22.0	72.4 ± 14.2	0.683
Soy protein (g/d)	8.2 ± 24.9	6.4 ± 8.5	0.030
Fat (g/d)	63.8 ± 26.1	61.8 ± 16.3	0.033
Carbohydrate (g/d)	235.2 ± 60.5	231.5 ± 49.5	0.127
Calcium (mg/d)	497.0 ± 266.9	531.0 ± 176.1	0.001
Vitamin D (IU/d)	100.4 ± 337.5	93.5 ± 245.0	0.594
Folate (μg/d)	221.0 ± 115.3	223.6 ± 62.1	0.520
Total lignans (μg/d)	414.5 ± 1197.6	525.1 ± 418.4	<0.001
MAT	4.1 ± 17.0	5.1 ± 5.5	0.060
LARI	165.5 ± 206.4	201.6 ± 104.6	<0.001
PINO	194.9 ± 916.1	258.5 ± 300.8	0.031
SECO	51.0 ± 112.0	60.7 ± 54.4	<0.001
Estimated ENL (μg/d)	313.5 ± 763.7	392.7 ± 277.9	<0.001
Dietary fibers	8.3 ± 14.4	9.6 ± 4.7	0.005

Dietary intake was assessed by validated food frequency questionnaires. The intakes of food groups and nutrients among case and control subjects were expressed as mean ± standard deviation and compared by independent *t*-test. * Nutrient intakes were estimated using the residual method, regressed by dietary total energy. ENL: enterolactone. MAT: matairesinol; SECO: secoisolariciresinol; PINO: pinoresinol; LARI: lariciresinol. Estimated enterolactone (ENL) = 0.62 × MAT + 0.72 × SECO + 1.01 × LARI + 0.55 × PINO.

**Table 3 nutrients-14-01100-t003:** Odds ratios (ORs) and 95% confidence intervals (CIs) of hip fracture risk according to quartiles of dietary total and individual lignans intakes in Chinese older adults.

	Quartiles(Q) of Total and Individual Lignans Intakes (Energy Adjusted)				P_trend_
	Q1	Q2	Q3	Q4	
Total lignans					
No. cases/controls	392/267	286/268	226/267	166/267	
Model 1 (crude OR)	1.00	0.708 (0.492, 1.017)	0.482(0.324, 0.718)	0.272(0.170, 0.434)	<0.001
Model 2 (fully adjusted OR)	1.00	0.477(0.284, 0.803)	0.353(0.186, 0.667)	0.237(0.103,0.544)	<0.001
Matairesinol (MAT)					
No. cases/controls	335/267	313/267	236/268	186/267	
Model 1 (crude OR)	1.00	0.856(0.593,1.236)	0.706(0.479, 1.039)	0.264(0.160, 0.436)	<0.001
Model 2 (fully adjusted OR)	1.00	0.827(0.487, 1.405)	0.531(0.299, 0.943)	0.222(0.104, 0.475)	<0.001
Lariciresinol (LARI)					
No. cases/controls	404/267	275/268	229/267	162/267	
Model 1 (crude OR)	1.00	0.516(0.355,0.748)	0.414(0.278, 0.619)	0.348(0.226, 0.535)	<0.001
Model 2 (fully adjusted OR)	1.00	0.427(0.253,0.721)	0.296(0.163,0.538)	0.374(0.199, 0.703)	<0.001
Pinoresinol (PINO)					
No. cases/controls	358/267	331/268	213/267	168/267	
Model 1 (crude OR)	1.00	0.900(0.619,1.309)	0.489(0.327,0.731)	0.241(0.147, 0.395)	<0.001
Model 2 (fully adjusted OR)	1.00	0.932(0.536,1.620)	0.346(0.189,0.632)	0.221(0.107, 0.460)	<0.001
Secoisolariciresinol (SECO)					
No. cases/controls	355/267	253/268	266/267	196/267	
Model 1 (crude OR)	1.00	0.844(0.576,1.236)	0.902(0.610,1.333)	0.535(0.353, 0.809)	<0.001
Model 2 (fully adjusted OR)	1.00	1.135(0.655,1.966)	0.771(0.428,1.389)	0.572(0.289, 1.134)	0.084
Enterolactone (ENL)					
No. cases/controls	467/267	277/268	189/267	137/267	
Model 1 (crude OR)	1.00	0.562(0.386, 0.816)	0.255(0.163, 0.398)	0.195(0.122, 0.313)	<0.001
Model 2 (fully adjusted OR)	1.00	0.540(0.325, 0.897)	0.318(0.173, 0.587)	0.224(0.116, 0.433)	<0.001

Data were analyzed by multivariate conditional logistic regression model (Cox regression) using the forward (Wald) method. Model 1 was the crude ORs and 95% CI. Model 2 was the full adjusted model with covariates including body mass index (kg/m^2^), education, incomes, medical history of fractures (yes or no), stroke (yes or no), coronary heart disease (yes or no), and hyperlipidemia (yes or no); current smoking (yes or no), regular tea-drinking (yes or no), regular calcium supplement (yes or no), body-weight-adjusted total physical activity (Mets/kg), dietary energy intake (kcal/d), energy-adjusted dietary fat (g/d), energy-adjusted dietary calcium (mg/d), energy-adjusted dietary soy protein (g/d), energy-adjusted dietary cholesterol (mg/d). Estimated enterolactone (ENL) = 0.62 × MAT + 0.72 × SECO + 1.01 × LARI + 0.55 × PINO.

**Table 4 nutrients-14-01100-t004:** Odds ratios (ORs) and 95% confidence intervals (95% CIs) of hip fracture risk according to quartiles of lignans from various food sources and dietary fibers in Chinese older adults.

	Quartiles(Q) of Lignan-Rich Food Intakes				P_trend_
	Q1	Q2	Q3	Q4	
Lignans from vegetables					
No. cases/controls	447/267	334/269	174/267	115/267	
Model 1 (crude OR)	1.00	0.432(0.279, 0.670)	0.251(0.161, 0.392)	0.180(0.110, 0.295)	<0.001
Model 2 (fully adjusted OR)	1.00	0.466(0.261, 0.835)	0.334(0.177, 0.629)	0.470(0.231, 0.954)	0.012
Lignans from nuts					
No. cases/controls	327/267	290/268	259/267	194/267	
Model 1 (crude OR)	1.00	0.761(0.531, 1.092)	0.634(0.426, 0.945)	0.296(0.180, 0.486)	<0.001
Model 2 (fully adjusted OR)	1.00	0.919(0.544, 1.551)	0.491(0.273, 0.881)	0.245(0.115, 0.521)	<0.001
Lignans from fruits					
No. cases/controls	396/267	360/268	193/268	121/267	
Model 1 (crude OR)	1.00	0.591(0.367, 0.951)	0.309(0.192, 0.496)	0.221(0.130, 0.376)	<0.001
Model 2 (fully adjusted OR)	1.00	0.804(0.417, 1.549)	0.866(0.437, 1.717)	0.756(0.356, 1.607)	0.612
Lignans from cereals					
No. cases/controls	252/267	273/268	273/268	272/267	
Model 1 (crude OR)	1.00	1.969(1.308, 2.964)	2.176(1.438, 3.292)	4.109(2.648,6.377)	<0.001
Model 2 (fully adjusted OR)	1.00	1.379(0.757, 2.512)	1.480(0.747, 2.933)	1.768(0.788, 3.968)	0.188
Dietary fibers					
No. cases/controls	491/269	292/267	183/268	104/267	
Model 1 (crude OR)	1.00	0.502(0.339, 0.742)	0.252(0.163, 0.389)	0.150(0.088, 0.254)	<0.001
Model 2 (fully adjusted OR)	1.00	0.661(0.405, 1.080)	0.420(0.238, 0.740)	0.375(0.194, 0.724)	0.001

Data were analyzed by multivariate conditional logistic regression model (Cox regression) using the forward (Wald) method. Model 1 was the crude ORs and 95% CI. Model 2 was the full adjusted model with covariates including gender, body mass index (kg/m^2^), education, incomes, medical history of fractures (yes or no), stroke (yes or no), coronary heart disease (yes or no), and hyperlipidemia (yes or no); current smoking (yes or no), regular tea-drinking (yes or no), regular calcium supplement (yes or no), body-weight-adjusted total physical activity (Mets/kg), dietary energy intake (kcal/d), energy-adjusted dietary fat (g/d), energy-adjusted dietary calcium (mg/d), energy-adjusted dietary soy protein (g/d), energy-adjusted dietary cholesterol (mg/d), dietary fish intake, and dietary milk and dairy products (g/d).

**Table 5 nutrients-14-01100-t005:** Odds ratios (ORs) and 95% confidence intervals (95% CIs) of hip fracture risk for quartiles of dietary total lignans intakes stratified by genders, source of controls, smoking, and alcohol drinking, etc. in Chinese older adults of Guangzhou, China.

Subgroup Analyses	Quartiles (Q) of Total Lignans Intake				P_trend_
	Q1	Q2	Q3	Q4	
Genders					0.085 *
Male					
No. cases/controls	55/29	77/59	82/77	111/63	
Adjusted OR (95%CI)	1.00	0.787(0.274, 2.261)	0.520(0.175, 1.545)	0.118(0.033, 0.431)	0.001
Female					
No. cases/controls	337/238	209/209	144/190	103/156	
Adjusted OR (95%CI)	1.00	0.358(0.201, 0.639)	0.263(0.135, 0.512)	0.252(0.112, 0.563)	<0.001
Source of control					<0.001 *
Community controls					
No. cases/controls	315/191	240/212	194/235	143/248	
Adjusted OR (95%CI)	1.00	0.274(0.123, 0.613)	0.242(0.104, 0.559)	0.109(0.040, 0.299)	<0.001
Hospital controls					
No. cases/controls	77/76	46/56	32/32	23/19	
Adjusted OR (95%CI)	1.00	0.826(0.410, 1.665)	0.464(0.157,1.371)	1.116(0.327, 3.812)	0.557
Body mass index (BMI)					0.659 *
BMI < 24					
No. cases/controls	291/179	214/161	185/163	126/156	
Adjusted OR (95%CI)	1.00	0.468(0.230, 0.953)	0.425(0.191, 0.950)	0.340(0.150, 0.768)	0.006
BMI ≥ 24					
No. cases/controls	96/86	68/106	40/104	38/111	
Adjusted OR (95%CI)	1.00	0.878(0.127, 6.045)	0.288(0.028, 2.999)	0.216(0.012, 3.782)	0.038
Healthy lifestyle scores					0.292 *
Scores < median					
No. cases/controls	258/122	160/107	113/75	72/64	
Adjusted OR (95%CI)	1.00	0.234(0.063, 0.866)	0.073(0.013, 0.422)	0.012(0.001, 0.200)	<0.001
Scores > median					
No. cases/controls	123/143	122/157	106/190	92/202	
Adjusted OR (95%CI)	1.00	0.354(0.140, 0.894)	0.195(0.068, 0.559)	0.223(0.071, 0.700)	0.004
Regular calcium supplementation					0.609 *
Yes					
No. cases/controls	122/111	93/108	49/127	49/108	
Adjusted OR (95%CI)	1.00	0.866(0.288, 2.609)	0.803(0.284, 2.271)	0.373(0.087, 1.608)	0.240
No					
No. cases/controls	270/156	193/160	177/140	117/159	
Adjusted OR (95%CI)	1.00	0.447(0.210, 0.954)	0.329(0.131, 0.827)	0.180(0.065, 0.498)	<0.001
Regular tea-drinking					0.576 *
Yes					
No. cases/controls	122/114	97/140	94/136	58/150	
Adjusted OR (95%CI)	1.00	0.425(0.134, 1.346)	0.680(0.244, 1.891)	0.213(0.056, 0.811)	0.056
No					
No. cases/controls	270/153	189/128	131/130	108/117	
Adjusted OR (95%CI)	1.00	0.284(0.121, 0.663)	0.086(0.026, 0.284)	0.178(0.054, 0.591)	<0.001
Sensitivity analysis					
Non-smoking					
No. cases/controls	346/247	240/233	168/232	135/222	
Adjusted OR (95%CI)	1.00	0.367(0.206, 0.652)	0.224(0.112, 0.451)	0.207(0.091, 0.468)	<0.001
No regular alcohol drinking					
No. cases/controls	373/255	269/250	204/248	150/239	
Adjusted OR (95%CI)	1.00	0.423(0.249, 0.720)	0.319(0.173, 0.589)	0.228(0.110, 0.470)	<0.001
No history of diabetes					
No. cases/controls	283/171	213/201	146/200	85/181	
Adjusted OR (95%CI)	1.00	0.417(0.234, 0.741)	0.258(0.134, 0.499)	0.222(0.104, 0.472)	<0.001

Data were analyzed by multivariate conditional logistic regression model (Cox regression) using the enter method. * Denoted *p* for an interaction that was derived by inclusion of a product term of a subgroup variable with quartiles of total lignans consumption; OR: odds ratio; 95%CI: 95% confidence interval; BMI: body mass index; Ca: calcium. Adjusted odds ratio in logistic regression models were adjusted for gender, age, education, incomes, medical history of fractures (yes or no), stroke (yes or no), coronary heart disease (yes or no), and hyperlipidemia (yes or no), current smoking (yes or no), regular tea-drinking (yes or no), body-weight-adjusted total physical activity (Mets/kg), dietary energy intake (kcal/d), energy-adjusted dietary fat (g/d), energy-adjusted dietary calcium (mg/d), energy-adjusted dietary soy protein (g/d), energy-adjusted dietary cholesterol (mg/d). For individual subgroup Cox-regression model data, the adjusted confounders would not include the variable of stratification.

## Data Availability

The original datasets are not publicly available due to ethical limitations publishing medical record data, but available from the corresponding author on reasonable request under strict confidential process.

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
