# Peer review of "Greater Consumption of Total and Individual Lignans and Dietary Fibers Were Significantly Associated with Lowered Risk of Hip Fracture—A 1:1 Matched Case–Control Study among Chinese Elderly Men and Women"

_nutrients, 2022, doi:10.3390/nu14051100_

Round 1

Reviewer 1 Report

The authors describe a novel and interesting case-control study that appears well-designed and is generally presented and discussed appropriately. Further suggestions for improving this manuscript are provided below.

  1. Line 51 - I suggest changing "are rich in" to "are found high amounts in [foods like]...". As written, the sentence suggests lignans contain high amounts of whole grains etc rather than the other way around.
  2. Line 67-68 - I suggest updating to "partly due to the assumption that typical Asian diets contains lower amounts of lignans" for clarity and to avoid repetition.
  3. Line 71 - replace "Till now" with "To date" o "Until now" or something similar to avoid use of colloquialism. The word "testifed" does not seem to be correct here. Should this be "tested" or "evaluated"?
  4. Lines 72-74 - rewrite this sentence for sense and clarity. As this appears to be of major importance in rationalising your stated Aims, I also suggest that authors support this statement with appropriate references. 
  5. Line 87 (and elsewhere) - I suggest using the term "older adults" rather than "elderlies".
  6. Line 97 - I suggest updating the start of the sentence to "A total of 1070 patients..." for clarity and sense.
  7. Line 302 - "grains" appears to be spelled incorrectly. Please update this.
  8. Discussion (possibly around line 350) - the authors should also briefly discuss limitations in relation to the available databases of lignans used. presumably there would be some challenges applying such data to an FFQ from a different country?

Author Response

Cover letter

Dear Ms. Li,

We would like to thank the reviewers for providing us with valuable comments and feedback on our manuscript entitled “Greater consumption of total and individual lignans and dietary fibers were significantly associated with lowered risk of hip fracture - A 1:1 matched case-control study among Chinese elderly men and women” (manuscript ID: nutrients-1617526). We have amended our manuscript accordingly using the Track Changes. Attached please find a point-by-point response to all of the comments raised by the reviewers.

We hope that the revised version and all the changes fulfill the requirements to make the manuscript acceptable for publication in Nutrients.

Thanks a lot for your consideration.

Yours sincerely,

Yu-ming Chen (Prof.)

School of Public Health,

Sun Yat-sen University (North Campus)

Office line: 862087330605

Response to Reviewers’ Comments

Reviewer 1

The authors describe a novel and interesting case-control study that appears well-designed and is generally presented and discussed appropriately. Further suggestions for improving this manuscript are provided below.

R: We appreciate the positive comment from the Reviewer.

  1. Line 51 - I suggest changing "are rich in" to "are found high amounts in [foods like]...". As written, the sentence suggests lignans contain high amounts of whole grains etc rather than the other way around.

R:  Thanks. We have revised.

  1. Line 67-68 - I suggest updating to "partly due to the assumption that typical Asian diets contains lower amounts of lignans" for clarity and to avoid repetition.

R: Thanks for the comments. As suggested, we revise the sentence as:

“Observational studies on lignans have been mainly conducted in European or American countries. Asians ingested less amount of dietary lignans than westerns (200~300 vs. >1000μg/d) and it is uncertain if such a low consumption is favorable for bone health[1].”

  1. Line 71 - replace "Till now" with "To date" or "Until now" or something similar to avoid use of colloquialism. The word "testifed" does not seem to be correct here. Should this be "tested" or "evaluated"?

R:  Thanks. The sentence has been revised as: “To date, no studies have evaluated the association of habitual lignans consumption…”

  1. Lines 72-74 - rewrite this sentence for sense and clarity. As this appears to be of major importance in rationalising your stated Aims, I also suggest that authors support this statement with appropriate references. 

R:  Thanks for the suggestion. We have revised the sentence and added one reference on the increased animal foods intake in China:

“In addition, China is in a transitional period from decades ago of plant foods dominated dietary pattern to current animal foods pattern[2]. It is uncertain whether such a low consumption of dietary lignans is favorable for bone health. “

  1. Line 87 (and elsewhere) - I suggest using the term "older adults" rather than "elderlies".

R: Thanks. In several places of the text, the “Chinese elderly” or “Asian elderly” has been revised as “Chinese older adults” or “Asian older adults”.

  1. Line 97 - I suggest updating the start of the sentence to "A total of 1070 patients..." for clarity and sense.

R: Thanks. Have changes.

  1. Line 302 - "grains" appears to be spelled incorrectly. Please update this.

R:  Have revised. Thanks.

  1. Discussion (possibly around line 350) - the authors should also briefly discuss limitations in relation to the available databases of lignans used. presumably there would be some challenges applying such data to an FFQ from a different country?

R. Thanks for the reviewer’s valuable comments. We have revised the discussion part accordingly as following:

“Another limitation could be that lignans databases we used in current study were from different countries since Chinese data were unavailable. The major lignans database in our analysis were from Japan where food varieties and dietary patterns were similar to those of Chinese population.”

Reviewer 2

This case-control study clearly and convincingly provides data to further our understanding of the association of dietary intake of lignans with the risk of hip fractures in a specific population, the Chinese elderly. The authors provide evidence with a 1:1 age and gender-matched case-control design and multicenter data collection that the greater consumption of lignans and lignans-rich foods were significantly associated with decreased risk of hip fracture for this population.

The major critic of this study is that the researchers did not take any biomarkers for the ligands during the study, however, this was addressed by the authors. I think that this manuscript  will contribute to our understanding and future analyses with respect to the association of nutrition and hip fracture in other populations.

R: We greatly appreciate the positive comments from the Reviewer.

References

  1. Zamora-Ros R, Knaze V, Lujan-Barroso L, Kuhnle GG, Mulligan AA, Touillaud M, Slimani N, Romieu I, Powell N, Tumino R et al: Dietary intakes and food sources of phytoestrogens in the European Prospective Investigation into Cancer and Nutrition (EPIC) 24-hour dietary recall cohort. European journal of clinical nutrition 2012, 66(8):932-941.
  2. He Y, Yang X, Xia J, Zhao L, Yang Y: Consumption of meat and dairy products in China: a review. The Proceedings of the Nutrition Society 2016, 75(3):385-391.

Reviewer 2 Report

This case-control study clearly and convincingly provides data to further our understanding of the association of dietary intake of lignans with the risk of hip fractures in a specific population, the Chinese elderly. The authors provide evidence with a 1:1 age and gender-matched case-control design and multicenter data collection that the greater consumption of lignans and lignans-rich foods were significantly associated with decreased risk of hip fracture for this population.

The major critic of this study is that the researchers did not take any biomarkers for the ligands during the study, however, this was addressed by the authors. I think that this manuscript  will contribute to our understanding and future analyses with respect to the association of nutrition and hip fracture in other populations.

Author Response

(The authors gave the same response as above.)
